# A Sensaptic ADAS Device Using Shape Memory Alloy Wires: Design and Control

**DOI:** 10.3390/ma14133494

**Published:** 2021-06-23

**Authors:** Deivamoney Josephine Selvarani Ruth, Kaliaperumal Dhanalakshmi, Seung-Bok Choi

**Affiliations:** 1Robert Bosch Centre for Cyber Physical System, Indian Institute of Science, Bangalore 560012, India; djsruth@gmail.com or; 2Department of Instrumentation and Control Engineering, National Institute of Technology, Tiruchirappalli 620015, India; dhanlak@nitt.edu; 3Department of Mechanical Engineering, The State University of New York, Korea (SUNY Korea), Incheon 21985, Korea

**Keywords:** shape memory alloy (SMA), pedal position sensing, haptics, synergistic, sensaptics, haptics, common rail system

## Abstract

This paper presents an active accelerator pedal system based on an integrated sensor and actuator using shape memory alloy (SMA) for speed control and to create haptics in the accelerator pedal. A device named sensaptics is developed with a pair of bi-functional SMA wires instrumented in a synergistic configuration function as an active sensor for positioning the accelerator pedal (pedal position sensing) to control the vehicle speed through electronic throttle and as a variable impedance actuator to generate active force (haptic) feedback to the driver. The reaction force emanated from the pedal alerts the driver and takes appropriate control action by slowing down the vehicle, in harmony with the road’s condition. The design is developed as a proof-of-concept device and is tested and evaluated in a real-time common rail diesel system for rail pressure regulation and over speeding tests, and the responses and performances are found to be promising.

## 1. Introduction

The World Health Organization (WHO) in 2018 issued statistics on global road safety, stating that annual road traffic deaths have reached 1.35 million. Traffic injuries are now the prominent killer for people aged between 5–29 years. The liability is inexplicably abided by walkers, cyclists, and motorcyclists, in developing countries [1]. Generally, crashes are attributed to driver factors, roadway factors, or vehicle factors, or are due to variable combinations of two or more of these factors. Though people are quite well aware of the general rules and safety measures while using roads, it is the laxity on their part which causes accidents and crashes. Some of the common erroneous behavior of humans which result in accidents are distractions to driver, over speeding, fatigue and sleeping, drunken driving, driving under the influence of drugs, red light jumping, avoiding safety gears like seat belts, non-adherence to lane driving, and overtaking in a wrong manner. The use of non-driving related devices such as music system, mobile phones, and more recently navigation systems, divert the driver’s attention away from safe driving. However, the driver has to pay continuous and constant attention to numerous visual stimuli like road, traffic signs, other vehicles, pedestrians, etc. which are growing in number all the time, in addition to the speedometer. Although technology enables to provide drivers with information that may be useful or even vital to the proper management of vehicles and traffic (imminent danger warnings, sleepiness alarms, etc.), this information also causes a distraction that generates other risks; consequently, driver’s inattention is considered as one of the major causes of road traffic accidents. However, most often accidents can be avoided if the driver can handle the threat, take appropriate decisions, and act within the crucial time, i.e., the intensity of the control action should be high to avoid a collision. Further, it is required that the controls should coexist; by the driver and a driving assistance system, both in coordination to enable smart driving, leading to a safe drive.

### 1.1. Driving Assistance Systems

Human life is most precious and hence, to ensure safe driving, car safety systems have evolved. To bond, the gap in improved passive safety systems like airbags and anti-crash bodies research has been focused more towards systems that actively support [2] include Anti-lock Braking Systems (ABS) [3,4], Electronic Brake-force Distribution (EBD), Advanced Cruise Control (ACC) systems, Lane Departure Warning Systems (LWDS) [5,6], and Vehicle Dynamics Control (VDC) [7], which actively regulates the vehicle dynamics demanded by the driving conditions. Although these systems do not explicitly help to improve driver’s ability to detect and respond to impending safety threats, they support improving the controllability and handling of cars, thereby enhance the driver’s capabilities to maintain the safe-field-of-travel and in an emergency. 

Intelligent Transport Systems (ITS) are focusing more on what [8] call response automation: “comfortably promoting safety”. Such systems like LDWS and Collision Warning Systems (CWS) automatically inform drivers when their safe-field-of-travel is degrading seriously, urging them to take swift action to restore this situation to an acceptable level of safety. The LDWS and CWS (warning systems) respectively warn drivers that they are deviating dangerously from their current lane and when their vehicle is getting dangerously close to the one in front, urging them to take corrective actions. The Advanced Driver Assistance System (ADAS) is a system that combines three functional entities: threat detecting sensors, force feedback actuators, and an algorithm that invokes the haptics, to help the driver for safe driving. When designed with such a safe Human–Machine Interface (HMI), it increases car safety and more generally road safety, but lacks the feature to automatically act at the appropriate instance. 

The contemporary and preferable need is an automatic driver support system that can take the necessary action by itself, without the command from the driver or the need for the driver to act, and demands an alert signal to the driver, in addition. The prime and primitive character of a driver assistance system is that it should be inherently pre-emptive. 

### 1.2. Haptic Devices for ADAS

Advanced driver assistance systems (ADAS) is an emerging technology to ensure safety and comfort travel for a safe drive. The types of information related to ADAS driving include spatial data, warning signals, communication data, coded information, and general information. Tactile warning signals were investigated for alerting drowsy drivers, notifying drivers of unplanned lane departures, and alerting drivers about possible imminent collisions [9] by interfacing with haptic human/machine interfaces. Vibrotactile patterns or signals that are chronologically and rapidly activated can give rise to the insight of the dynamic threat signal at different locations on the body in the seat [10], seatbelt [11], foot pedal [12], steering wheel [13], and the driver’s torso [14,15,16]. The performance of multisensory warning signals using both tactile and audio channels was also examined and found that the response for a tactile warning signal is fast, and a detailed study can be found in [17,18].

### 1.3. Motivation and Focus of This Work

While electric motors or pneumatic elements are capable of performing haptics, why not shape memory alloy (SMA), which is commonly known only as an actuating element? The characteristic of SMA that makes it suitable for haptics is its thermo-mechanical behavior to develop an alert signal by functioning as an actuator. Apart from being only a normal/simple actuator, it can operate as a stiffness varying element (variable impedance actuator) by which it can provide an ideal HMI. The SMA-based support system proposed in this work senses the accelerator pedal position as the input command to control the speed playing the role of the position sensor. Besides, the SMA wire acts as a Variable Impedance Actuator (VIA) by varying its stiffness concerning the threat index, such as obstructing vehicles, pedestrians, and animals, and by force feedback warns the driver, and most importantly takes needed control action of slowing down the vehicle in response to the intensity of the threat. The test results on Active Accelerator Pedal (AAP) for its speed regulation are obtained and analyzed in an automotive test rig for varied driving conditions. The organization of the paper starts with a brief description of sensaptic pedal, its design and development for ADAS, control implementation with feedback, the tests on the diesel rail system, followed by the test results.

## 2. Sensaptic Pedal for ADAS

Haptics is certainly the most interesting, as it exploits a sensory channel that is not yet overloaded. The benefits of using haptic information are the high speed of processing and reflexive action to threat information, compared to the same visual/audio information. It was therefore decided to use haptics as the best way to alert the active support information to drivers. Variable Impedance Actuators (VIAs) are designed to be physical human–machine interaction actuators that change their stiffness during operation or adapt stiffness according to situations. VIAs are capable of restraining impedance in terms of stiffness and/or damping along with force/torque continuously and independently, as disparate to classical actuators, and are preferable to be used in active driver assistance systems. This haptic information is brought about by a newly proposed SMA-based device named by the sensaptics technique which is engaged with the accelerator pedal as an integral part of the primary input devices to provide haptic information to the driver. These are inherent properties in SMA and can be designed as variable impedance actuators due to its generation of high actuation density in a small volume compared to the classical electrical actuators, and also to recover these strains when a load/temperature is removed. 

A SMA based sensaptic device is developed to function as the sensor (continuously) and actuator (conditionally), as shown in Figure 1. The SMA sensaptic device consists of two units in synergetic configuration; two SMA wires are connected/wound in open rectangular loops between pairs of discs D_A1_ (fixed) and D_A2_ configured to be the outer unit A, and D_B1_ and D_B2_, and the inner unit B. The displacement gain of a single unit (∆L/L) is named after the units as A for the outer unit and A+B for the inner unit, as it reflects the additive strains of both the units which are described in detail in [19]. The (active) sensing ability of SMA in the device is availed favorably to sense the pedal position in the way to meet the demanded torque to maintain the desired speed. SMA actuator is known for its self-sensing property, which allows it to be used as an active sensor in its actuation. The change in resistance in SMA due to load (input force on the pedal, applied by the driver) induced in the martensite phase is used for the pedal position sensing.

This hybrid sensaptic device is fixed to the accelerator pedal of a car to perform as sensor and actuator, non-autonomously within a shared workspace, and also to physically interact with the driver establishing a human–machine interface. The physical AAP is connected to the engine testbed through a Continental Software Module, a specialized industrial tool that is used for testing automotive components. It is noted here that the technical contribution of this work is to use a single device featuring both sensing and actuation functions, while the classical one use independent devices to perform both the functions.

## 3. Dynamics of Active Accelerator Pedal 

The states of the foot and the accelerator pedal are time-dependent dynamical systems. A stationary condition, the driving force on the pedal, must be equal to the pedal force on the driver’s foot. A simple mechanical can be modeled for the foot and pedal as mass, spring, and damper system, with external forces acting on it to produce changes in foot and pedal position. The pedal dynamics [20] are shown in Figure 2. To yield a deflection $\alpha_{P}$ of the pedal, the driver must apply a force *F_D_* on it, which is linearly linked to the dynamic properties of the pedal–stiffness *Ks*, damping *B_P_*, mass *M_P_*, and variable impedance actuator *K_VIA_*. Due to the accelerator return spring, the pedal returns as soon as the foot pressure is removed or reduced. The dynamic equation of the pedal is given by
(1)$$F_{D}\left(t\right)=\;M_{P}\frac{d^{2}\alpha_{P}\left(t\right)}{dt^{2}}+B_{P}\frac{d\alpha_{P}\left(t\right)}{dt}+K_{S}\alpha_{P}\left(t\right)$$

For the fixed base pedal module, the mass and damping are negligibly small compared to the stiffness of the pedal (return) spring, reducing to passive spring. This spring, however, is pre-tensed, and to produce a pedal deflection an initial force *F_i_* has to be applied.
(2)$$F_{D}\left(t\right)=K_{S}\alpha_{P}\left(t\right)+F_{i}$$

The function of the return spring is installed to pull or push back the pedal to its normal zero deflection position. At zero deflection there is no input to the engine; at maximum pedal deflection, input to the engine is maximal. The design of the AAP includes the pedal and the pre-tensed passive return spring attached to the pedal lever. The range of depression angle and pedal movement is limited by a mechanical stopper. 

The reduced dynamic equation of motion of the AAP is given by
(3)$$F_{D}\left(t\right)=(K_{S}+K_{VIA\;}\left(t\right))\alpha_{P}\left(t\right)+F_{i}$$

The stiffness of the AAP is
(4)$$K_{AAP}=K_{S}+K_{VIA}(t)$$
where,
(5)$$K_{P}=\frac{F_{D}}{\alpha_{P}}$$
(6)$$K_{S}=\frac{F_{D}}{y}$$
(7)$$K_{VIA}\left(t\right)=\frac{F_{D}-F{\left(i\right)}_{SMA}}{x-x_{A}}$$
where *y* is the elongation of the pedal spring, *x* is the displacement of the sensaptics when not under actuation, and *x_A_* is the displacement of the sensaptics under actuation. *K_P_* is the stiffness of the accelerator pedal, *K_VIA_* is the stiffness of the VIA in the sensaptics, and *F*(*i*)*_SMA_* is the force generated corresponding to an activation current *i*, during actuation. The stiffness of the VIA in the sensaptics *K_VIA_* contributes to additive stiffness of the pedal in the AAP, hence aids the speed of deceleration. In AAP, the sensaptics is activated when a threat is encountered and, it exerts opposing haptic force to be felt by the driver. 

The AAP is put to use in the SDAS for haptic control. While the sensaptics is in actuation, its force and/or stiffness variation can be used for haptic feedback. In AAP, force feedback is realized by the sudden generation of force by the current input to the wires according to the intensity of threat (initial force acts as haptics), and its stiffness is varied by varying the current.

### 3.1. Force Feedback

In a force feedback system, an additional force *F_VIA_*(*t*) is imposed directly with the inherent characteristics of the accelerator pedal. The mechanical equivalent of the AAP dynamics with force feedback is shown in Figure 3a; its dynamic equation of motion becomes
(8)$$F_{D}\left(t\right)=\;K_{S}\alpha_{P}\left(t\right)+F_{i}+F_{VIA}\left(t\right)\;$$

The above dynamics are presented in Figure 3b for *F**_VIA_*(*t*) = 0 and when *F_VIA_*(*t*) > 0; it is seen that the force–position characteristic for *F_VIA_*(*t*) = 0 is translated upwards from the other. This means that, for any given pedal position, the force required to hold that position is increased with the magnitude *F_VIA_*(*t*).

The force exerted on the normal pedal is about 20 N, while the AAP generates a force of 30 N, for no torque or speed demand. The slope of the force–position characteristic of the AAP remains the same as that of the normal pedal i.e., an amount α_P_ is needed to change the pedal position and pedal force (Δ*F_D_*) is required compared to the nominal force–position characteristic of the AAP at *F_VIA_*(*t*) = 0. Changes in AAP force can be perceived only when the force *F_VIA_*(*t*) is changed to enable the haptics.

### 3.2. Stiffness Feedback

In Stiffness feedback, the dynamic feedback system of Equation (8) does not increase the pedal force for any given pedal position, but it increases the pedal force depending on these three parameters of pedal position, velocity, and acceleration. The only parameter with which information can be transferred effectively is the added stiffness *K_VIA_*. The added stiffness can be felt as an increase in the pedal force $F_{VIA}$(t) = $K_{VIA}$(t)$\;\alpha_{P}$(*t*) when the pedal is in a static position, as seen in Figure 4a,b, which shows the position–force relationship of AAP. 

As an additional benefit, a change in pedal position Δ$\alpha_{P}\left(t\right)$ now also yields an increase in pedal force (Δ*F_D_*(*t*) = Δ*α_P_*(*t*) (*K_S_* + *K_VIA_*(*t*)) that is bigger than the increase in pedal force for the nominal pedal characteristic (Δ*F_D_*(*t*) = *K_S_*Δ*α_P_*(*t*)), since the slope of the force–position characteristic is increased from *K_S_* to *K_S_* + *K_VIA_*(*t*). Then the dynamic equation of the AAP with stiffness feedback (including the pre-tensed spring) becomes
(9)$$F_{D}\left(t\right)=\;(K_{S}+K_{VIA\;}\left(t\right))\alpha_{P}\left(t\right)+F_{i}$$

The stiffness of the normal pedal and AAP are 1.5 and 2.5 N/m, respectively.

## 4. Smart Driver Assistance System (SDAS)

Drivers are capable of acting in their environment through their accelerator pedal, but the information they use to base these actions is mainly acquired visually, by looking through the windscreen as shown in Figure 5a which is the longitudinal control of the car. The haptic AAP interface is an in-vehicle device that does not inform the driver about the state of the vehicle itself, but about the state of the vehicle in the environment. The drivers would be able to determine the boundaries of their safe-field-of-travel as a mental image from the haptic information of the field boundaries, besides vision as shown in Figure 5b.

With a force, *F_D_*, the displacement of the pedal, *α_P_*, determines the speed and longitudinal position of the car. The visual information allows the driver to determine the required amount of accelerator pedal press to maintain the desired safe-field-of-travel, closing the control loop. The haptic information is accessible continuously throughout the control, the control actions can be felt too at the pedal, as they will induce changes in the safe-field-of-travel. The Smart Driver Assistance System (SDAS) senses both the relative speed and distance of the threat as an input. This information is communicated to the driver through the AAP by haptics signal in terms of an upward force *F_VIA,_* in the accelerator pedal, which is actively exerting the force on their foot. 

The roles of sensaptics in the driver assistance system are summarized as follows: (1) Sensing the pedal position of the car by capturing the electrical resistance variation of SMA, for any applied force on the pedal. (2) Controlling the dynamics of the car and displaying it to humans through haptic feedback and takes control action by varying the pedal impedance. (3) Using the pedal assistive forces to improve the kinematic response of the driver’s behavior.

### Haptic Feedback in the Control Loop

Haptics by force and stiffness feedback from SMA based sensaptics are implemented for SDAS is a closed-loop. The driving simulator is designed in Virtual Reality Modeling Language (VRML) with the dynamics of the car and controller for the sensaptics integrated into MATLAB^®^. VRML and MATLAB^®^ both perform interactive simulations. The sequence of data transfer between just the driver and the assistance is depicted in Figure 6a. Figure 6b illustrates the interaction between the driver in a virtual environment to check the proposed active accelerator pedal sensaptics.

An intelligent control algorithm is essential for the haptic system. To make an intelligent system, it should be capable to make decisions based on its inputs. Fuzzy–PID controller is used to control the current through the SMA in the outer unit of sensaptics by variation in its stiffness, to bring about haptics feel in the AAP. This controller is structurally simple and exhibits robust performance over a wide range of operating conditions. A proportional-integral-derivative controller (PID) is a simple basic control but is not sufficient for a nonlinear system. However, fuzzy control provides a convenient and formal methodology with human heuristic knowledge. Therefore, the advantages of fuzzy and PID control are incorporated into a controller to achieve high control performance for nonlinear systems. The fuzzy logic controller is designed to tune the desired current, which is the input to the PID controller framed by fuzzy control rules.

The rules are framed based on the knowledge about the system with the following conditions: (1) The voltage corresponding to the minimum and maximum position of the pedal is 0 and 5 V. (2) A distance of 10 m between the test vehicle and conflict object is considered as the safe-field-of-travel in Virtual reality machine language (VRML) environment (the pedal will provide haptic feedback when the relative distance is less than 10 m). (3) Speed of the vehicle is monitored continuously. When the relative speed is more and the corresponding relative distance is less than 10 m, the stiffness is varied accordingly. (4) If the pedal position is zero, the output will be zero for all cases. It is noted here that we have given the threshold valve of 10 m relative distance to evoke the SDAS module for the threat avoidance. Thus, when the relative distance is less than are equal to 10 m the stiffness of the AAP device varies in this case. Therefore, the SMA in the sensaptics module will actuate which will give an upward force (haptics), and an upward force in the accelerator pedal will reduce the speed of the vehicle. A combination of Fuzzy–PID controller is designed with pedal position, the relative distance between the threat and the test car, and speed of the test vehicle as the input variables. The output of the fuzzy is the desired current that would be able to maintain the desired stiffness in the pedal to the intensity of the threat. The fuzzy output offers the essential desired current, from the previous empirical knowledge of the AAP, and is tuned for optimal performance. The fuzzy controller uses the error (e) and the rate of change of error ($\dot{e}$) as its inputs and meets the desired tuning parameters based on time-varying. The representation of the fuzzy–PID control is shown in Figure 7. The rules and membership functions of input and output variables of the system are shown in Figure 8 and Figure 9, respectively. Figure 8 describes the range of the rules pedal position, the distance between the vehicles, the speed of the vehicles, and the current to the actuator in the sensaptics. The DAS offers technology that alerts the driver to potential problems like active car following, lane change, and collision avoidance, and takes corrective action automatically. The controller is implemented for the said control actions in the SDAS and tested in the VRML environment.

## 5. Diesel System Test Facility

This section briefly presents the standard test bench facility at an Automotive Company in India used for performing special tests for diesel systems. The test bench facilitates the running of specific tests for diesel engines for any suitable performance testing. The electronic control unit connects the hardware with a harnessed wiring. The ECU has software with variables and control algorithms. Since the bench is primarily for testing the diesel fuel injection systems, the other components are either masked its functionality or simulated to run as a system. The interface between software to the outside world is built with a specific tool from ETAS GmbH (Engineering Tools, Application and Services), known as INCA (INtegrated Calibration and Acquisition tool). INCA provides a platform for human and diesel systems helps in calibration, diagnostics, and validation of the systems and components. The software for the electronic control of the system is intelligently developed as functional modules from acquiring the engine speed and position of crank and camshaft, acquiring the driver demand by way of accelerator pedal signal, controlling the diesel fuel pressure at the fuel rail, injection into combustion chamber, air supply system management, and interfacing function to improve the efficiency of after-treatment controls.

The tests on the bench are carried out with a simulated electrical engine. The actual hardware of the active accelerator pedal is used to feed the driver demand to the ECU. As the accelerator pedal module also contains a force feedback provision, the feedback of this force feedback is wired directly to INCA to capture its parameters at the synchronized time with other engine system parameters. A special test case was designed to validate the functionality of the active accelerator pedal sensor and force feedback unit on the diesel system test bench, and the arrangement is set up as in Figure 10.

### 5.1. Test Run for Sensing Performance of the Sensaptics in AAP

The physical AAP is connected to the engine testbed through a Continental Software Module, the software is a specialized industrial tool that is used for testing automotive components. In this experiment, as the pedal is pressed, the position sensor from the pedal sends the pedal position information to the software module, and based on the pedal position the required torque demand is calibrated and sets the desired rail pressure to meet the demand. An internal control logic increases the current rail pressure of the system to meet the desired rail pressure, thereby meeting the torque demand of the driver.

The working of sensaptics as the APP is evaluated experimentally for the desired rail pressure at a constant speed. Constant pedal position is related to steady-state vehicle speed. The driver’s requirement for more or less torque is a key input variable for controlling the air charge electronically. The accelerator-pedal module provides this variable as a sensor signal. The input to the common rail fuel injection system of the diesel engine in the testbed is the pedal position, which is from the APP, the potentiometer. As well, the sensing signal from the sensaptics as the APP is measured for comparison. The pedal is depressed from its idle position (0% throttle) to the maximum position (100% throttle) and then slowly released. The valves are correspondingly controlled by the ECU such that it attains the desired rail pressure with constant speed. Figure 11a depicts the flow diagram of rail pressure regulation on a common rail diesel system. Figure 11b showcases the flow diagram of the real-time working operation of the sensaptic in-loop with the common rail diesel test rig.

Electronic (engine) control unit (ECU) controls the engine with respect to the airflow and the fuel to have the combustion, so we have two sections in the Throttle position sensor (TPS), which is the resistance valve from the pedal position which will be fed as the input through Electronic throttle valve (ETV) and correspondingly the valve will be opened for the airflow through the ECU. The pressure regulation valve is the one to control the pressure in the fuel flow or the speed of the fuel as an input into the engine for combustion. This is again regulated by the regulated pressure senor, and it is fed to the ECU.

Figure 12 shows the test results achieved by implementing the proposed sensaptics. Figure 12a depicts the force acting on the pedal, Figure 12b shows the respective change in electrical resistance of the SMA in the sensaptics, calibrated in terms of voltage. The sensing performance of sensaptics is compared with the standard potentiometer in Figure 12c; the performance plot depicts that the sensaptics faithfully reproduces the pedal position and fulfills the function of pedal position sensing, genuinely like the potentiometer. Figure 12d depicts the profile of the torque demanded to meet the desired rail pressure; the monitored rail pressure is shown in Figure 12e,f, and shows the test speed outline.

### 5.2. Test Run for Actuating Performance of the Sensaptics

To examine the haptics performance of the sensaptics in the AAP, the control algorithm is modified to indicate the over-speeding to the driver. The speed of the vehicle is due to the deflection of the pedal, i.e., the pedal position corresponds to the speed. The correct pedal position is estimated from the speed information drivers get visually. Visual cues from the road and peripheral stationary objects might indicate absolute speed. For exact knowledge of the absolute speed, the speed indicator is available. Though there is a speed indicator, the driver has to necessarily observe it to know the exact speed of the vehicle. Alternatively, this AAP would communicate to the driver at the pedal by haptics when over-speeding. The pedal self serves as an indicator of Overspeed. As the speed of the engine crosses a set value, the haptic actuation is activated to alert the driver at the pedal; a predefined speed (1300 rpm) is set as the maximum speed limit.

Figure 13 depicts the results of the experiment conducted to alert over speeding. Figure 13a shows the position of the pedal varied from minimal to maximal deflection. The corresponding change in the engine speed is shown in Figure 13b. The force acting on the pedal corresponding to the change in pedal position is shown in Figure 13c. It is observed from Figure 13b that at 25 s the engine speed crosses the maximum limit of 1300 rpm and from Figure 13c shows a significant change in the magnitude of force required to press the pedal, to meet the actual speed. This is an observation of overspeeding, and the additional force on the pedal is the alert signal to the driver. Within the speed limit the maximum force required to press the pedal to its maximum position is about 30 N, as shown in Figure 13a. However, when the set speed of the engine is crossed, the haptic actuation system is activated, which makes the driver apply nearly 80 N of force to push the pedal to its maximum position. This distinct difference in force can easily be realized by the driver through haptics in the accelerator pedal. Figure 13d,e shows the torque demand and the rail pressure of the engine, which is set by the sensaptics to control the electronic throttle valve.

## 6. Conclusions

A new technique is called shared sensing and actuation, wherein the SMA wires do the function of both sensing and actuation to display its bi-functionality of the element. The structure is called sensaptics, an active accelerator pedal system, based on an integrated sensor and actuator technique for advanced driving assistance systems using SMA for speed control and to create haptics in the accelerator pedal. The proof of concept is developed, and it is tested in the standard automobile testbed for the overspeeding and pressure regulation tests. The tests prove that the concept and the principle of operation are standard to drive a diesel engine, and AAP system design needs to be improved in terms of handling the momentum of the vehicle and to be customary for successful real vehicle application in a practical environment.

## Figures and Tables

**Figure 1 materials-14-03494-f001:**
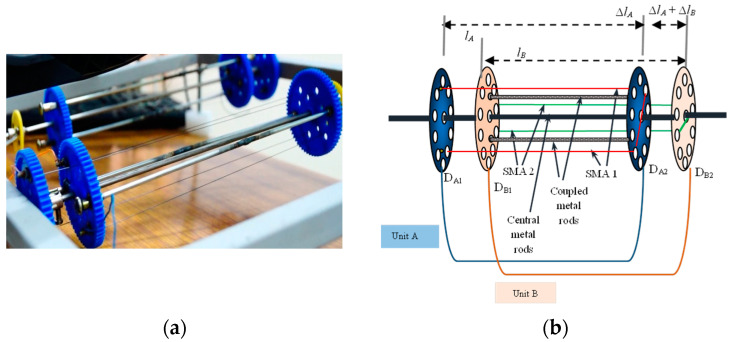
SMA based sensaptic device; (**a**) photograph of the device, (**b**) schematics of the model with labeling.

**Figure 2 materials-14-03494-f002:**
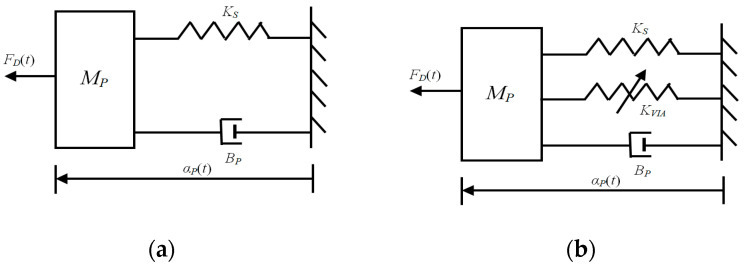
Mechanical model equivalent of the pedal dynamics; (**a**) accelerator pedal, (**b**) active accelerator pedal.

**Figure 3 materials-14-03494-f003:**
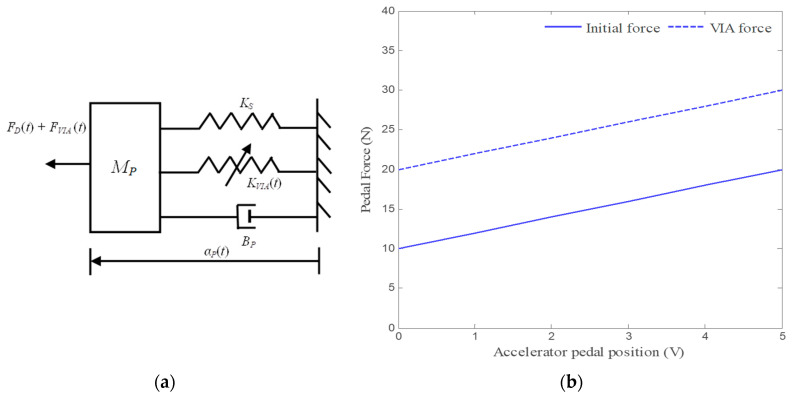
Mechanical model and characteristics; (**a**) schematic mechanical equivalent of the AAP dynamics with force feedback, (**b**) position–force relationship of the AAP.

**Figure 4 materials-14-03494-f004:**
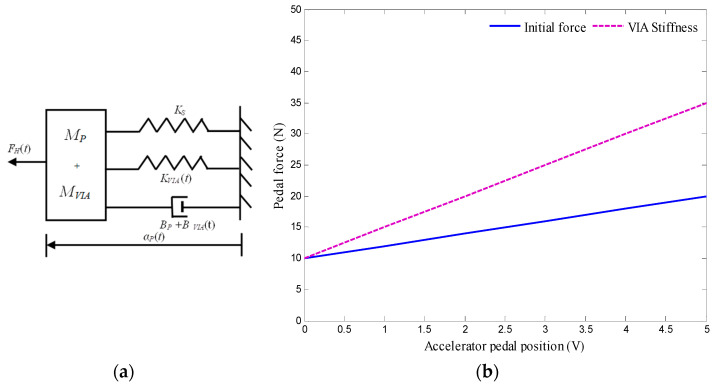
Mechanical model and characteristic; (**a**) schematic mechanical equivalent of the AAP dynamics with stiffness feedback, (**b**) position–force relationship of AAP with stiffness due to VIA.

**Figure 5 materials-14-03494-f005:**
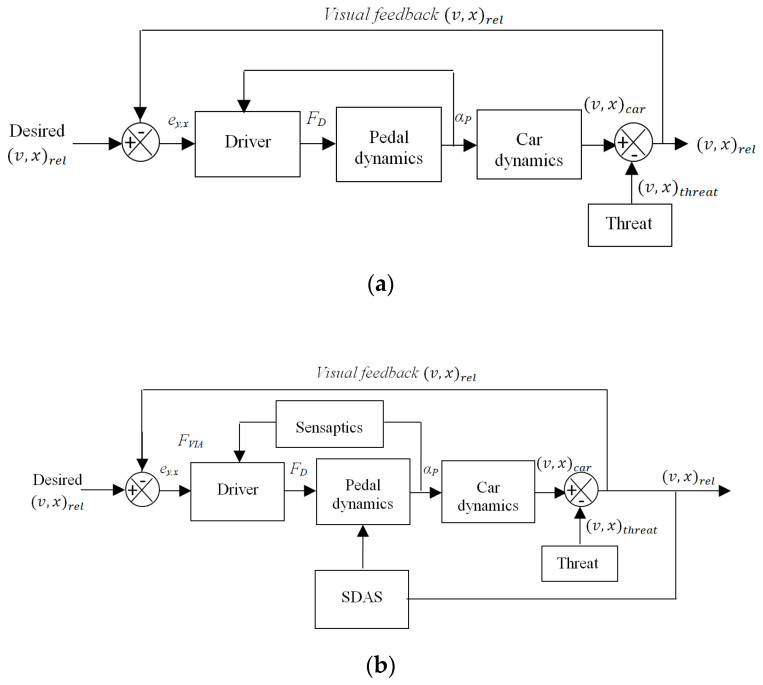
Control block diagram; (**a**) longitudinal control of the car, (**b**) longitudinal control with haptic feedback.

**Figure 6 materials-14-03494-f006:**
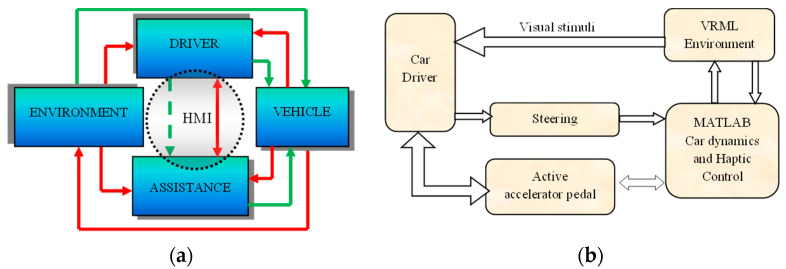
Flow of control signal for human–machine interaction; (**a**) schematic configuration, (**b**) control sequence.

**Figure 7 materials-14-03494-f007:**
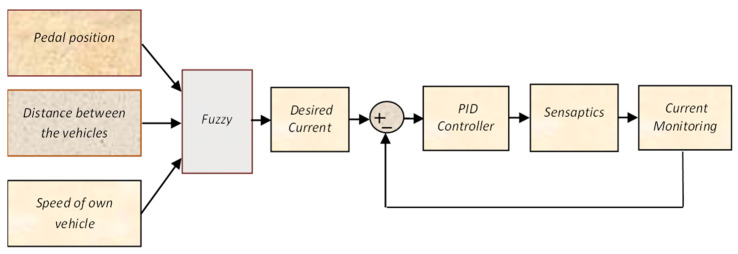
Schematic of the controller for the SDAS.

**Figure 8 materials-14-03494-f008:**
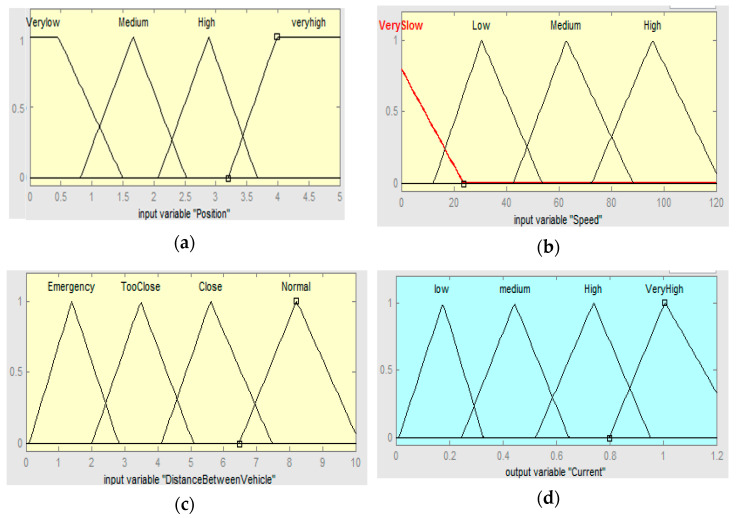
Membership functions of inputs and output; (**a**) pedal position, (**b**) distance between the vehicles, (**c**) speed of the vehicles, (**d**) output membership.

**Figure 9 materials-14-03494-f009:**
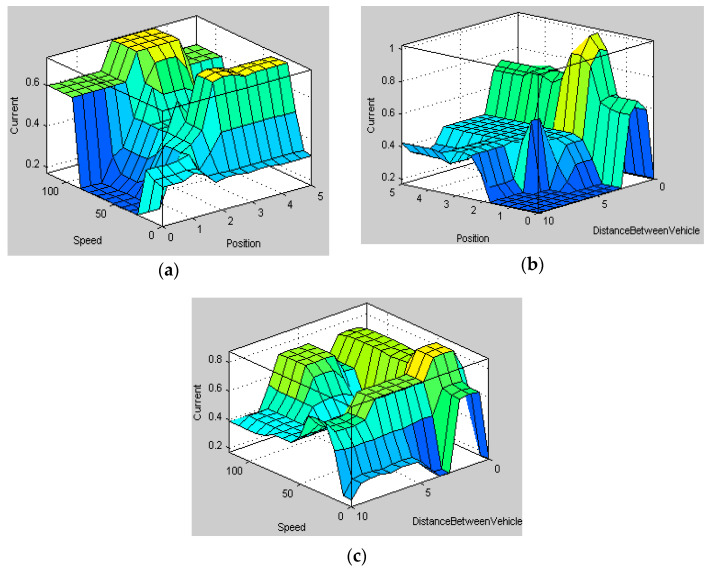
Rules for all the input parameters; (**a**) pedal position, (**b**) distance between the vehicles, (**c**) speed of the vehicles.

**Figure 10 materials-14-03494-f010:**
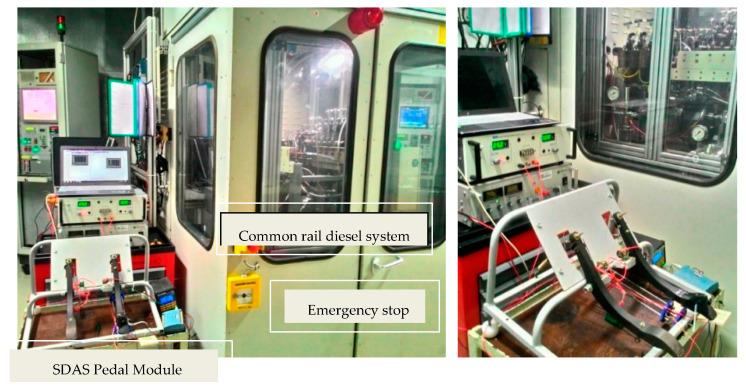
Photographs of the sensaptics under test run in common rail diesel system.

**Figure 11 materials-14-03494-f011:**
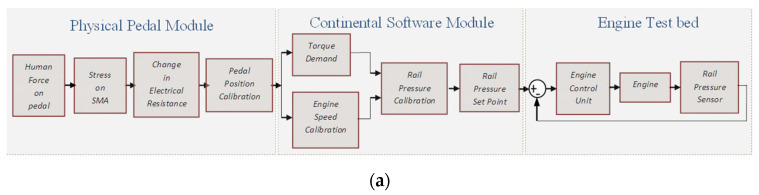
Test control block-diagram; (**a**) testbed at Continental Automotive Components (India) Pvt. Ltd., (**b**) flow diagram of rail pressure regulation on common rail diesel system.

**Figure 12 materials-14-03494-f012:**
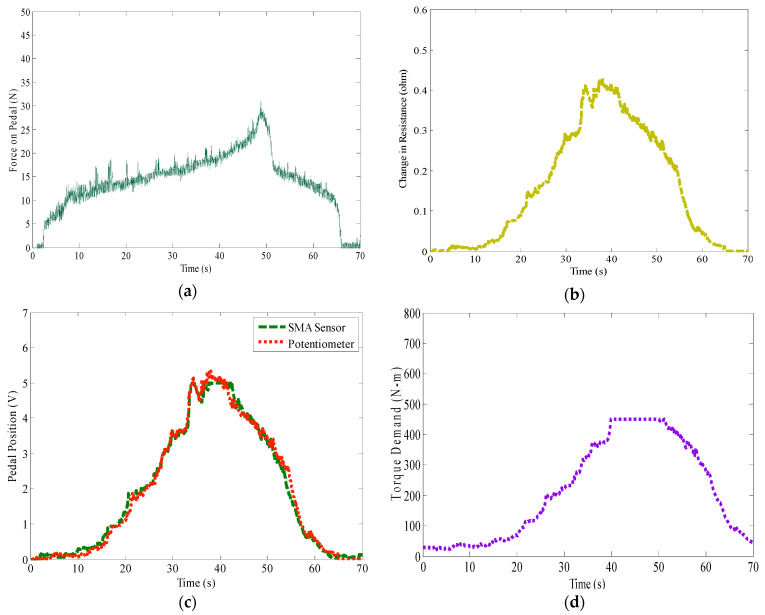
Test results of rail pressure regulation on common rail diesel system by the sensaptics; (**a**) force acting on AAP, (**b**) electrical resistance variation in SMA, (**c**) sensing signal of sensaptics as APP, (**d**) torque demand, (**e**) rail pressure—desired and actual, (**f**) test speed.

**Figure 13 materials-14-03494-f013:**
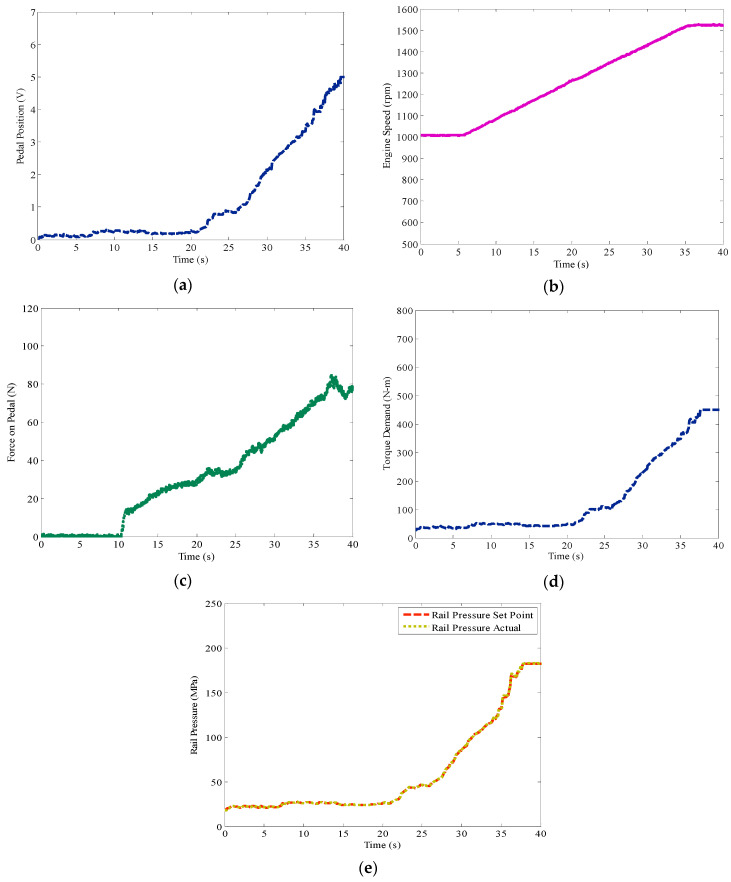
Test results of over speeding on common rail diesel system by the sensaptics; (**a**) sensing signal of AAP, (**b**) engine speed, (**c**) force acting on AAP, (**d**) torque demand, (**e**) rail pressure—desired and actual.

## Data Availability

The data presented in this study are available on request from the corresponding author.

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
