# Peer review of "A Sensaptic ADAS Device Using Shape Memory Alloy Wires: Design and Control"

_materials, 2021, doi:10.3390/ma14133494_

Round 1

Reviewer 1 Report

The paper presents an innovative use of shape memory alloys for driver assistance. The designed device combines the functions of environment reaction, driver information, and actuation. The paper integrates the development of control schemes and the results of experiments using a diesel test bench. The article is worthy of being published, with some following remarks.

The introduction is well written and the contribution is well presented and described.

In section 2.

It would be interesting to compare a little more the developments of this article with the reference [28], preliminary work by the authors (a priori the developments concern control and tests?).

In figure 1b), you have "SMA" and "SMA1"; these labels could be more explicit, e.g. "SMA1" and “SMA2".

In section 3.

In figure 2 and equation 1, you have "Ks" and in the text above: "Kp".

For equation 8, could you give the literal expression of Fvia?

Figure 4’s legend: you can correct "FFigure 4".

In section 3.1 you have stiffness feedback because Kvia depends on t in figure 3... What is then the difference between sections 3.1 and 3.2? How do you switch technologically from force feedback to stiffness feedback?

In section 4.

Figure 7: the fuzzy block is not linked to the PID parameters, although this tuning is mentioned in the text. Why?

Figure 8: letter c) is missing under the corresponding diagram.

Could you give in the text an analysis of figures 8 and 9?

In section 5.

Clarify some of the terms in figure 11: TPS, ETV, RPS, PRV.

I have difficulty interpreting Figure 12f in relation to the other figures: why is the velocity constant from the start?

Could you give a value for the variation of the slope at 25 s for figure 13c?

Other remarks.

I would recommend adding some technical elements about the sensaptic device centered on SMAs (describe a little more the architecture of the device shown in figure 1):

- Why 4 SMA wires?

- Why are they arranged in the presented way?

- How the device works?

- What about the wiring?

- How is the resistance of the wires measured?

Reviewer 2 Report

The authors present an active accelerator pedal system based on an integrated sensor and actuator technique that can be integrated in advanced driving assistance systems using shape memory alloy for the purpose of speed control and to create haptics in the accelerator pedal.

Dear Authors, please consider the following comments:

  1. General Comments
  • The article needs a general review of English language and style, in particular the introduction section.
  • The manuscript is difficult to read and follow. The structure and relevance of the content presented must be carefully reviewed to ensure a clear presentation of the proposed system. It is recommended to carefully review, update and shorten the introduction section, focusing the text on the topic covered. A methodology section with presentation of the main stages of the study and their interconnection must be included in the manuscript in order to improve the understanding of the information presented through the text. Section 5 also needs revision to improve the clarity of the "Diesel System Test Facility" description and test performed.
  • Figures must be presented in the text before they appear. Some figures need a more detailed description, for example: explain in detail figure 5 (b), especially the differences and gains in relation to figure 5 (a); explain figures 6 and 11 (b); it is not possible to clearly interpret what is represented in figure 10.
  • The conclusions section should be improved, namely by clearly presenting the main benefits of using the proposed system, its limitations and future developments.
  1. Specific comments:
  • Abstract needs careful revision. The gap / novelty addressed must be clearly presented, as well as the main results obtained and expected from the large-scale application of the proposed system.
  • Keywords must be reconsidered and must be limited to the main ones.
  • The introduction section should include more recent references. Examples: pag.1 - use the most recent WHO information available (the ones presented in the manuscript are almost 20 years old); the same for the information cited from the IRF and presented in section 1.1 Driving Assistance Systems. Citations and references related to information obtained from WHO and IRF must be included in the text and references section.
  • Page 1, last line: the citation [1] does not match “Department of Environment, Transport and the Regions (DETR) (2000)”.
  • Page 3, end of section 1.1: since the authors include section "1.4. Motivation and focus of this work", the objectives of the work, as well as the manuscript organization description must be included in this section. Avoid repetition of the work objective throughout the text.
  • The content of section 1.2 is confusing (page 3) and its real relevance to the developed system is unclear. Review and consider including some of this information in section 1.3, eliminating section 1.2.
  • Only on page 7 and 10 it is possible to understand what kind of threat can trigger and how the proposed SDAS (AAP) works. This information must be presented at the beginning of the manuscript. Review this aspect.
  • Page 9, section 4. Smart Driver Assistance System: the first two sentences of this section repeat information previously presented. Consider eliminating these sentences. According to what is presented in this section, regarding reference [29], the proposed solution is not innovative. Clarify this aspect by presenting the novelty introduced in the proposed system in relation to existing / studied similar systems.
  • Last paragraph on page 11, "(3) Speed of the vehicle is monitored continuously. When the relative speed is more and corresponding relative distance is less than 10m, the stiffness is varied accordingly." Why the authors consider a 10m relative distance regardless the vehicle speed? Note that the perception-reaction time of drivers is approximately 2 seconds, representing different travelled distances depending on the vehicle speed.

Reviewer 3 Report

The paper entitled “ A Sensaptic ADAS Device Using Shape Memory Alloy Wires: Design and Control“ can be quite interesting and informative for the readers of the journal. It is presenting an interesting low-cost solution with promising results of tests carried out in a real-time automotive test bench.

However, I have the following comments that hopefully help the authors improve their paper:

The majority of the information provided in the Introduction of paper is obsolete. E.g. information on estimation of annual traffic monetary loss is from 2012, information about WHO road traffic injuries rank is from 2003, information about the percentage of human contributory factors (driver-errors) to traffic accidents are from 2000. Situation related to traffic accidents changed significantly during the last decade. There were many efforts made to decrease the number of road accidents, also the infrastructure, vehicle park and applied technologies changed a lot. For all reasons presented it is essential to use the most recent information available.

Also, the list of active driving assistance systems is missing complexity. I recommend that the authors consider using the structure that is presenting these systems in a more comprehensible form (https://doi.org/10.1051/matecconf/201710700024).

Page 5 – The first paragraph that is beginning with the sentence „Literature shows...“ should be appropriately referenced (provide sources that are confirming its content).

The quality of some figures should be improved (at least figure 1(b), figures 3(b), 4(b), graphics provided in figure 8 and schemes provided in figure 11).

Please correct the line spacing of the text provided in the paragraph after figure 1.

Please correct the text „Fig. 4 FFigure 4.“

Conclusions section is quite short. This part of  paper should be elaborated in more detail. Were identified any weaknesses of proposed AAP module? What are the main limitations for the practical implementation of presented pedal model?

Were any similar solutions identified throw the literature review? It can be very supportive to compare the results of experiments with other approaches. The main contribution of this paper compared to the available literature needs to be better outlined.

What are the directions of further research? What are the implications of the presented solution for theory and practice?

I recommend English revision and simplification of the text -  especially in the abstract - several sentences are too complicated and the formulations are not fully comprehensible.

Round 2

Reviewer 2 Report

Authors present an active accelerator pedal system based on an integrated sensor and actuator technique that can be integrated in advanced driving assistance systems using shape memory alloy for the purpose of speed control and to create haptics in the accelerator pedal.

Dear authors,

The effort made to improve the article is visible. The article is more focused on the subject covered and easier to follow.

However, responses to reviewer comments were ambiguous and the changes made were not adequately explained. Consider improving this aspect in the future.

Some small aspects of English style should still be refined (please check one more time the text). For example, in the abstract sentence "A pair of bi-functional SMA wires instrumented in a synergistic configuration function as an active sensor for positioning the pedal accelerator (pedal position sensing) to control the vehicle speed through electronic throttle and as a variable impedance actuator to generate active force (haptic) feedback to the driver; the design is a proof-of-concept." Apparently something is missing from the sentence or the link to "the design is a proof-of-concept" part is missing.

Also in the abstract, the acronym SMA must be included in parentheses the first time it appears written in full. In keywords SMA must be included in full and in parentheses (“shape memory alloy (SMA)”).

The main results obtained in the diesel system test are still missing from the abstract, namely in terms of sensing and actuating performance.

At the end of section 1.3 "Motivation and focus of this work" is still missing a description of the article organization - following sections.

Section 2 is an important section of the work that should demonstrate that authors are aware of what has already been done in the area of "sensaptic pedal for ADAS". It lacks significant bibliographic support.

Section 3, page 5: Where does "Kp" appear in the formulation prior to expression 5?

As mentioned in the previous review, images must appear positioned in the manuscript after being presented and explained in the text. This aspect has not been verified in the revised version. Text between lines 223-233 on page 6 should appear before Figure 3. Text between lines 371-285 on pages 10 and 11 should appear before Figures 7, 8 and 9. Text between lines 494-502 on page 14 should appear before Figure 12.

Section 4, Page 9, line 339: the meaning of "PID" must be included in full.

Section 4, Page 9: it remains unclear why the authors adopted a distance of 10m regardless of vehicle speed. Authors must support this value with bibliographic references or evidence. How the system is triggered from the defined distance is understandable, what is not understood is why the value used is 10m (why not 20 or 30m?).

The explanation of Figure 10 remains confusing. The images are small and the caption very incomplete. This aspect should be frankly improved. Has this equipment used in other SDAS/ADAS tests/studies? Include this aspect in the text.

Figure 11 b): ETV meaning is missing.

The conclusions continue to require attention in order to clearly and sustainably present the findings of the work carried out and presented in the manuscript.

Formatting aspects:

- Use the same font and font size throughout the manuscript.

- Justify the entire text.

- Standardize the format of all diagrams and graphics presented in the manuscript.

- Check all figures captions format.

Reviewer 3 Report

The quality of the paper improved. The authors adapted it according to the comments. However, there are still few shortcomings.

In explanation of parameter ap (used in formula 1) it is recommendable to use the same symbol as in the formula.

There should be provided an explanation of the parameters used in Figure 1. Also the description of the parameter KVIA used in figure 2 is missing.

All parameters used in the text of the paper should be explained consistently. I also recommend to provide at least basic description of the forces and processes that are shown on figures. These aspects of the paper should be improved.

I am not sure if the quality of some figures is enough good.

After revising the listed aspects I will be happy to recommend this paper for publication.

Round 3

Reviewer 2 Report

Dear authors,
Despite being a non-native English speaker, I still have reservations about the English used in the sentences added in order to respond to the reviewers' suggestions.
The location of the explanatory text of the figures and figures still needs revision (although the authors refer that this operation was carried out, the revised version made available to reviewers maintains, in most cases, the previous structure). These aspects must be carefully reviewed by the authors to ensure clarity.
Bibliographical framework of the adopted approach and the adoption of a distance of 10m for the system's triggering remains unclear.

Reviewer 3 Report

All the comments had been taken into consideration by the authors and the manuscript was adapted accordingly. Its quality improved significantly , so I am happy to recommend it for publication.
